# The Prognostic Role of BRD4 Expression in High-Grade Serous Ovarian Cancer

**DOI:** 10.3390/cancers16111962

**Published:** 2024-05-22

**Authors:** Angeliki Andrikopoulou, Garyfalia Bletsa, Angeliki Rouvalis, Dimitris Tsakogiannis, Maria Kaparelou, Alkistis Papatheodoridi, Dimitrios Haidopoulos, Michalis Liontos, Meletios-Athanasios Dimopoulos, Flora Zagouri

**Affiliations:** 1Department of Clinical Therapeutics, Alexandra Hospital, National and Kapodistrian University of Athens, 11528 Athens, Greece; kapareloum@gmail.com (M.K.); alkipapath@med.uoa.gr (A.P.); mlionto@med.uoa.gr (M.L.); mdimop@med.uoa.gr (M.-A.D.); fzagouri@med.uoa.gr (F.Z.); 2Research Center, Hellenic Anticancer Institute, 10680 Athens, Greece; bletsag@yahoo.gr (G.B.); dtsakogiannis@gmail.com (D.T.); 3Obstetrics and Gynecology, 1st Obstetrics and Gynecology Clinic, National and Kapodistrian University of Athens, 10509 Athens, Greece; angrouvalis@hotmail.com (A.R.); dchaidop@med.uoa.gr (D.H.)

**Keywords:** BRD4 expression, serous, ovarian cancer, BET proteins, BET inhibitors, prognosis

## Abstract

**Simple Summary:**

BET proteins (BRD2, BRD3, BRD4 and BRDT) serve as the readers of DNA acetylation and regulate the expression of key genes in cancer. BET inhibitors were recently developed and have shown efficacy in Phase I/II clinical trials in ovarian cancer. The most thoroughly studied BET protein is BRD4, which is amplified in ~18–19% of ovarian cancer cases according to *The Cancer Genome Atlas* (TCGA). We explored the prognostic role of the BRD4 gene and protein expression in advanced high-grade serous ovarian carcinoma (HGSC). Low BRD4 gene expression was associated with shorter overall survival compared to intermediate/high BRD4 gene expression at 12 months (95% CI; 1.75–30.49; *p* = 0.008). Low BRD4 gene expression was also associated with worse progression-free survival (PFS) both at 12 months (95% CI; 1.2–16.5; *p* = 0.03) and at 24 months (95% CI; 1.1–8.6; *p* = 0.048). There was no significant association between BRD4 protein expression and overall survival.

**Abstract:**

Background: Bromodomain and extra-terminal (BET) domain proteins that bind to acetylated lysine residues of histones serve as the “readers” of DNA acetylation. BRD4 is the most thoroughly studied member of the BET family and regulates the expression of key oncogenes. BRD4 gene amplification has been identified in ovarian cancer (~18–19%) according to *The Cancer Genome Atlas* (TCGA) analysis. BET inhibitors are novel small molecules that displace BET proteins from acetylated histones and are currently tested in Phase I/II trials. We here aim to explore the prognostic role of the BRD4 gene and protein expression in the ascitic fluid of patients with advanced FIGO III/IV high-grade serous ovarian carcinoma (HGSC). Methods: Ascitic fluid was obtained from 28 patients with advanced stage (FIGO III/IV) HGSC through diagnostic/therapeutic paracentesis or laparoscopy before the initiation of chemotherapy. An amount of ~200 mL of ascitic fluid was collected from each patient and peripheral blood mononuclear cells (*PBMCs*) were isolated. Each sample was evaluated for BRD4 and GAPDH gene expression through RT-qPCR and BRD4 protein levels through enzyme-linked immunosorbent assay (ELISA). The study protocol was approved by the Institutional Review Board of Alexandra University Hospital and the Committee on Ethics and Good Practice (CEGP) of the National and Kapodistrian University of Athens (NKUA). Results: Low BRD4 gene expression was associated with worse prognosis at 12 months compared to intermediate/high expression (95% CI; 1.75–30.49; *p* = 0.008). The same association was observed at 24 months although this association was not statistically significant (95% CI; 0.96–9.2; *p* = 0.065). Progression-free survival was shorter in patients with low BRD4 gene expression at 12 months (5.6 months; 95% CI; 2.6–8.6) compared to intermediate/high expression (9.8 months; 95% CI; 8.3–11.3) (95% CI; 1.2–16.5; *p* = 0.03). The same association was confirmed at 24 months (6.9 months vs. 13.1 months) (95% CI; 1.1–8.6; *p* = 0.048). There was a trend for worse prognosis in patients with high BRD4 protein levels versus intermediate/low BRD4 protein expression both at 12 months (9.8 months vs. 7.6 months; *p* = 0.3) and at 24 months (14.2 months vs. 16.6 months; *p* = 0.56) although not statistically significant. Again, there was a trend for shorter PFS in patients with high BRD4 protein expression although not statistically significant both at 12 months (*p* = 0.29) and at 24 months (*p* = 0.47). Conclusions: There are contradictory data in the literature over the prognostic role of BRD4 gene expression in solid tumors. In our study, intermediate/high BRD4 gene expression was associated with a favorable prognosis in terms of overall survival and progression-free survival compared to low BRD4 gene expression.

## 1. Introduction

Ovarian cancer is the seventh most common malignancy in women accounting for 324,484 new cases in 2022 [1]. Unfortunately, the incidence continues to rise, with an estimated 47% increase in new cases by 2045 [2]. High-grade serous ovarian carcinoma (HGSC) is the most common histology and comprises the 70% of epithelial ovarian cancer (EOC). HGSC is a member of Type II ovarian cancer that harbors mutations in tumor suppressor gene TP53 and BRCA1/2 genes. TP53 mutations is an almost universal phenomenon in HGSC affecting up to 96% of these tumors [3]. Germline BRCA1/2 mutations are identified in up to 15% of serous ovarian cancer, while an additional 5–7% of ovarian cancers harbor somatic BRCA1/2 mutations [3]. *The Cancer Genome Atlas* (TCGA) reported that around 50% of HGSC tumors carry other mutations in genes associated with homologous recombination repair (HRR) [3]. Mutations in HRR genes such as ATM, CHEK2, RAD51, CHEK2 and PALB2 lead to homologous recombination deficiency (HRD) in HGSC and offer a druggable target. HRD tumors comprise the 50% of HGSC and benefit from treatment with poly (ADP-ribose) polymerase (*PARP*) inhibitors. PARP inhibitors were first approved in December 2014 for pre-treated advanced HGSC with germline BRCA-mutations and were subsequently approved for maintenance treatment after first-line chemotherapy in 2018 [4,5]. Since the introduction of PARP inhibitors in clinical practice the prognosis of HGSC has changed dramatically. The mutational landscape of high-grade ovarian cancer implicates other gene mutations as well (RB1, NF1, FAT3 and CDK12), although with no therapeutic significance [3]. There is an unmet need to identify other genetic alterations that could be potentially targeted in HGSC. 

Epigenetic modifications were first described by Conard Waddington in 1942 [6]. These changes include the alterations that modify DNA function without affecting DNA sequence. Epigenetic modifications rely on three major mechanisms: (1) histone regulation, (2) DNA methylation and (3) non-coding RNAs. Histone acetylation at the ε-amino group of lysine residues in H3 and H4 tails protruding from the histone core is one of the main epigenetic modifications involved in cancer. This process is mediated by three different groups of proteins: histone acetyltransferases (HATs) that transfer acetyl groups to histone lysine residues, histone deacetylases (HDACs) that remove them and bromodomain and extra-terminal (BET) domain proteins which serve as the “readers” of DNA acetylation [7]. The BET family consists of four members (BRD2, BRD3, BRD4 and BRDT) that share a common architecture comprising two N-terminal bromodomains (BD1 and BD2) which interact with acetylated lysine residues and a more divergent extraterminal (ET) structure at the C-terminal [7]. BRD4 is the main BET protein associated with cancer and serves as a transcriptional regulator that controls gene expression of key oncogenes such as MYC. BRD4 gene amplification has been identified in ovarian cancer (~18–19%) according to *The Cancer Genome Atlas* (TCGA) analysis while increased BRD4 mRNA levels have been identified in 9% of ovarian cancers [8]. BRD4 accumulates to promoters and super-enhancer regions of target genes and interacts directly with acetylated transcription factors, including RelA, ERα, p53 and TWIST [8]. BET inhibitors (BETis) are small molecules that target BET proteins and displace BRD4 out of super-enhancer regions. The antitumor activity of BET inhibitors was first described in 2008 [9]. Since then, a number of BET inhibitors with different properties entered preclinical and clinical trials including JQ1, I-BET151, ABBV-075, I-BET762 and OTX015 [9]. It is now clear that BRD4 could serve as a novel target in ovarian cancer like BRCA1/2 mutations and HR deficiency.

We have previously described in detail the rationale and clinical role of BET inhibition in ovarian cancer [10]. We here aim to identify BRD4 gene expression and protein levels in patients with advanced FIGO stIII/IV HGSC. BRD4 gene and protein expression will be subsequently associated with prognosis in ovarian cancer. 

## 2. Materials

Ascitic fluid was obtained from 28 patients with HGSC through diagnostic/therapeutic paracentesis or laparoscopy performed at Alexandra University Hospital. All patients originated from Greece and received treatment in the Oncology Department of Alexandra University Hospital. Fluid collection was performed before the initiation of chemotherapy and all patients were diagnosed at an advanced stage (FIGO III/IV). All patients provided informed consent to participate in the study. The study protocol was approved by the Institutional Review Board of Alexandra University Hospital (Protocol Number: 725/14-10-2020) and the Committee on Ethics and Good Practice (CEGP) of the National and Kapodistrian University of Athens (NKUA). The study was performed in compliance with the Declaration of Helsinki.

## 3. Methods

Ascitic fluid was obtained after diagnostic/therapeutic paracentesis or laparoscopy before the initiation of treatment. An amount of ~200 mL of ascitic fluid was collected from each patient. Peripheral blood mononuclear cells (*PBMCs)* were *isolated* from ascitic fluid by *Ficoll* density gradient centrifugation and cryopreserved at −80 °C for subsequent testing. Each sample was further evaluated for BRD4 and GAPDH gene expression through RT-qPCR and BRD4 protein levels through enzyme-linked immunosorbent assay (ELISA). RNA was extracted from ascitic cells using KIT NucleoSpin RNA Blood (Macherey-Nagel GmbH, Düren, Germany) according to the manufacturer’s instructions. RNA was subsequently quantified using a Qubit™ Flex Fluorometer (Invitrogen Ltd., Life Technologies, Renfrew, UK) and Qubit™ RNA BR Assay kit (Invitrogen Ltd., Thermo Fisher Scientific, Oxford, UK). For RNA quality control, samples were loaded into agarose gel electrophoresis 1.2% and then stained with Midori Green Direct (Nipon Genetics, Europe GmbH, Düren, Germany). RNA integrity with no DNA contamination was displayed as two distinct bands at 4500 bp and 2000 bp corresponding to 28s και 18s rRNA. In case of DNA interference, DNA was then cleaved by DNAse I (Invitrogen Ltd., Thermo Fisher Scientific, UK) according to the manufacturer’s instructions. The samples were then gain loaded in a 1.2% agarose gel to confirm the DNA removal. Subsequently, RT-qPCR was performed to quantify gene expression of the target genes.

### 3.1. One-Step RT-qPCR

One-step reverse transcription quantitative polymerase chain reaction (RT-qPCR) was *performed* to *measure* the *expression* of the *BRD4 gene.* The *expression* of housekeeping protein *GAPDH* was *used* as loading *control.*

Primers used are listed in the table below:
**Gene****Primer****Sequence 5′→3′****References**BRD4BRD4_FCCATGGACATGAGCACAATCWu T et al., 2015 [11]BRD4_RTGGAGAACATCAATCGGACAGAPDHGAPDH_FTTCACCACCATGGAGAAGGCGAPDH_RCCCTTTTGGCTCCACCCT

Cobas Z 480 (Roche Diagnostics Ltd., Rotkreuz, Switzerland) was used for the detection of BRD4 and GAPDH gene transcripts. Reverse transcription and real-time polymerase chain reaction (PCR) was performed through KAPA SYBR^®^ FAST One-Step qRT-PCR Master Mix (2X) (KAPA BIOSYSTEMS, Wilmington, MA, USA) that allows the conversion of mRNA to *complementary DNA (cDNA*) and the amplification of cDNA. One-step RT-qPCR was performed with a 10 μL final volume. Every reaction included 75 ng of total RNA, 200 nM of each primer, 5 μL of reaction buffer (KAPA SYBR FAST qPCR Master Mix 2Χ), 0.2 mM dUTP and ddH_2_O to 10 μL final volume. Reverse transcription of target mRNA was performed at 42 °C for 5 min. cDNA incubation was performed at 95 °C for 3 min and then 40 cycles of attenuation at 95 °C for 10 s, 40 cycles of hybridization at 60 °C for 20 s and elongation at 72 °C for 1 s. Finally, a melting curve was produced from the amplified products ranging from 65 °C to 97 °C. Data collection was at 72 °C in 510 nm according to the manufacturer’s instructions.

The identification of the primers’ efficiency was confirmed through one-step RT-qPCR using 1:10 serial *dilutions of total RNA* (75 ng–0.075 ng). The efficiency of each primer was 100%. The template used for the analysis was DCT: ΔCt = Ct Brd4 − Ct GAPDH, RQ = 2^−ΔCt^ × 100%. All the tests were performed three times. 

For one-step RT-qPCR reaction:
**Reaction Kit****Concentration**Primer (Forward) 10 pmol200 nMPrimer (Reverse) 10 pmol200 nMdUTP (10 mM)0.2 mMKAPA SYBR FAST qPCR Master Mix 2Χ1×50X KAPA RT Mix1×RNA template75 ngddH2OUp to 10 μL

### 3.2. Protein Isolation from Ascitic Cancer Cells

Protein extraction from ascitic samples was conducted with lysis buffer Cell Extraction Buffer (Invitrogen Ltd., Life Technologies, UK) according to the manufacturer’s instructions. Protease inhibitor reagent (Sigma–Aldrich) was also applied. Samples were centrifuged at 800 rpm and cells were washed out twice with PBS. Subsequently, the supernatant was removed and 5 mL of frozen PBS was applied and centrifuged at 800 rpm. 

### 3.3. Protein Quantification with ELISA

For the identification of BRD4 protein, KIT Human BRD4 (Bromodomain-containing protein 4 Elisa Kit) (Assay Genie, London, UK) was used. All measurements were repeated three times. Multiskan™ FC Microplate Photometer (Thermo Fisher Scientific, Inc., Waltham, MA, USA) was used to measure the fluorescence intensity.

## 4. Statistical Analysis

BRD4 levels were stratified in three groups according to low, intermediate and high gene expression. Overall survival (OS) was defined as the time between the start of chemotherapy and the date of death from any cause. Progression-free survival (PFS) was defined as the time between the start of chemotherapy and the date of disease progression. Kaplan–Meier estimates were used to describe and visualize the effect of categorical variables on OS and PFS. Log-rank tests were used to explore the prognostic value of categorical variables on clinical outcomes. Clinicopathological characteristics including age at diagnosis, histology, disease stage, type of surgery performed (primary versus secondary), debulking status, type of chemotherapy administered, maintenance treatment, lines of treatment, disease-free survival and overall survival were collected from patient files and were registered on an electronic database. All statistical analysis was performed using SPSS (version 23) (SPSS, Inc., Chicago, IL, USA). 

## 5. Results

### 5.1. Clinicopathological Characteristics

Clinicopathological characteristics are summarized in Table 1. The median age at diagnosis was 66 years (42–86) and all patients were diagnosed with HGSC stage FIGO IIIC. A total of 36% (10/28) of the patients underwent primary debulking surgery (PDS) while the majority of patients (43%; 12/28) underwent interval debulking surgery (IDS). Performance status was ECOG 0–1 (89%; 25/28) in most cases and only three patients had performance status ECOG 2. BRCA1/2 mutation status was available in all patients and HRD status in most of them. BRCA1/2 mutations were not identified in 78% (22/28) of the patients; BRCA1 mutations were present in 18% (5/28) of the cases while there was also one BRCA2 mutation. All patients received front-line treatment with paclitaxel and carboplatin. Most patients (64%; 18/28) received maintenance treatment with PARP inhibitors after the completion of neo-/adjuvant platinum-based chemotherapy while 43% (12/28) received maintenance treatment with bevacizumab either as monotherapy or in combination with PARP inhibitor Olaparib according to the positive results of phase 3 trial PAOLA-1 [12]. The median PFS on first-line chemotherapy was 11 months (1–33). Most patients recurred with peritoneal implants (71%; 20/28), lymph nodes (39%; 11/28) or ascites, pleural effusion, etc. 

### 5.2. BRD4 Gene Expression and Overall Survival (OS)

Patients were categorized in three subgroups according to BRD4 gene expression: low expression (<0.130), intermediate expression (0.130–0.354) and high expression (0.354–1.56). At 12 months, patients (*n* = 9) with low BRD4 gene expression had a median OS of 6.3 months (95% CI; 3.3–9.3), while patients with intermediate/high BRD4 expression had a median OS of 10.4 months (95% CI; 9.0–11.8). These data are illustrated in Table 2. Low BRD4 gene expression was associated with worse prognosis at 12 months compared to intermediate/high expression (95% CI; 1.75–30.49; *p* = 0.008). Kaplan–Meier curves of the two populations are illustrated in Figure 1.

We performed the same analysis of overall survival at 24 months. In patients with low BRD4 gene expression (*n* = 9) the median OS at two years was 10.3 months (95% CI; 3.3–17.3) versus 18.5 months (95% CI; 14.8–22.2) in patients with intermediate/high BRD4 expression. Low BRD4 gene expression was associated with short OS at 24 months (Figure 2) although this association was not statistically significant (95% CI; 0.96–9.2; *p* = 0.065). 

### 5.3. BRD4 Gene Expression and Progression-Free Survival (PFS)

Overall, there were 8 patients with low BRD4 gene expression and 17 patients with intermediate/high BRD4 gene expression among the 25 patients with PFS data. Low BRD4 gene expression was associated with shorter PFS at 12 months (5.6 months; 95% CI; 2.6–8.6) compared to intermediate/high expression (9.8 months; 95% CI; 8.3–11.3). Kaplan–Meier curves of PFS at 12 months are illustrated in Figure 3. The association of low BRD4 expression with worse PFS at 12 months was statistically significant (95% CI; 1.2–16.5; *p =* 0.03).

We performed the same analysis at 24 months. Similarly, low BRD4 gene expression was associated with shorter PFS at 24 months (6.9 months; CI; 2.1–11.7) compared to intermediate/high expression (13.1 months (95% CI; 10.0–16.3). This association was also statistically significant (95% CI; 1.1–8.6; *p =* 0.048). Kaplan–Meier curves of PFS at 24 months are illustrated in Figure 4.

### 5.4. BRD4 Protein Expression and Overall Survival

We then performed the same analysis according to BRD4 protein level as measured through ELISA. The median BRD4 protein expression was 201.4 pg/mL. Patients were divided into three different subgroups: low < 102 pg/mL, intermediate (102–323 pg/mL) and high BRD4 protein level (>323–1298 pg/mL). At 12 months, BRD4 protein level was not associated with survival in each subgroup (Figure 5). However, we noticed that the high BRD4 protein level was associated with worse survival at 12 months (7.6 months; 95% CI; 4.3–10.8) compared to both intermediate BRD4 protein expression (10.3 months; 95% CI; 8.2–12.4) and low BRD4 protein expression (9.3 months; 95% CI; 7.2–11.5). Therefore, we divided patients into two distinct subgroups: those with high BRD4 protein expression in ELISA assay versus those with intermediate/low BRD4 protein expression.

The median OS at 12 months was 9.8 months (95% CI; 8.3–11.4) in patients with intermediate/low BRD4 protein expression versus 7.6 months (95% CI; 4.3–10.8) in those with high BRD4 protein levels. There was a trend for worse prognosis in patients with high BRD4 protein expression although not statistically significant (*p* = 0.3) (Figure 6). 

We performed the same analysis for OS at two years. The median OS at two years was 14.2 months (95% CI; 6.4–22.1) for patients with high BRD4 protein expression compared to 16.6 months (95% CI; 12.9–20.4) in those with intermediate/low BRD4 protein expression (*p* = 0.56). Again, the association between BRD4 protein level and survival was not statistically significant (Figure 7). 

### 5.5. BRD4 Protein Expression and Progression-Free Survival

We then investigated the association between BRD4 protein level and PFS at one and two years from diagnosis. At 12 months, the median PFS was 6.9 months (95% CI; 3.5–10.3) in patients with high BRD4 protein expression compared to 9.2 months (95% CI; 7.6–10.9) in those with intermediate/low BRD4 protein expression. There was a trend for shorter PFS in patients with high BRD4 protein expression although not statistically significant (*p* = 0.29) (Figure 8). 

We performed the same analysis for PFS at two years. The median PFS was 9.5 months (95% CI; 3.6–15.4) in patients with high BRD4 protein expression compared to 11.9 months (95% CI; 8.7–15.1) in patients with intermediate/low BRD4 protein expression (*p* = 0.47) (Figure 9).

### 5.6. Association between BRD4 Gene and Protein Expression

We then investigated the relationship between BRD4 gene expression and protein levels. There was no linear association between the BRD4 gene and protein as illustrated in Figure 10 (*p* = 0.065). 

## 6. Discussion

Given the pivotal role of BRD4 in HGSC and the recent development of BET inhibitors we investigated BRD4 gene and protein expression in cells isolated from the ascitic fluid of patients with advanced disease. BET inhibitors have been shown to attenuate proliferation of ovarian cancer cells by various mechanisms: (1) inhibition of oncogenic pathways like FoxM1 and JAK/STAT, (2) resensitization of resistant cells to platinum agents and PARP inhibitors, (3) dysregulation of oxidative stress response and glycolysis, (4) disruption of homologous recombination (HR) capacity in BRCA wild-type cells and others [13,14,15]. The development of these new agents creates the imminent need to better characterize the role of the BRD4 gene in high-grade ovarian carcinoma. We aimed to associate BRD4 gene and protein expression with prognosis in patients with advanced HGSC. Intermediate/high BRD4 gene expression was associated with better one-year overall survival compared to low expression (10.4 months (95% CI; 9.0–11.8) versus 6.3 months (95% CI; 3.3–9.3)) (*p* = 0.008). The same trend was noted at two-year survival although not statistically significant (18.5 months (95% CI; 14.8–22.2) versus 10 months (95% CI; 3.3–17.3)) (*p* = 0.065). In terms of PFS, intermediate/high BRD4 gene expression was associated with a favorable prognosis both at one-year analysis (9.8 versus 5.6 months; *p* = 0.03) and at two years (13.1 versus 6.9 months; *p* = 0.048). In contrast, high BRD4 protein expression was associated with an adverse prognosis at one-year survival (*p* = 0.3) and at two-year survival analysis (*p* = 0.56), although not significantly. Of note, BRD4 gene expression had no linear correlation with BRD4 protein level.

There are data that link high BRD4 gene expression to a more favorable prognosis or to no prognostic significance in solid tumors [16,17]. In contrast, other studies have demonstrated an inverse association between BRD4 gene expression and survival in HGSC [8,18,19,20]. These contradictory results may emerge from multiple factors. Firstly, BRD4 amplification is usually identified in BRCA wild-type tumors [19]. Indeed, BRCA1/2 mutations are identified in 15–17% of HGSC tumors and are mutually exclusive with CCNE1 mutations, which are found in 30% of HR-proficient tumors [19,21]. The BRCA mutant-like profile is mainly characterized by gene alterations on chromosomes 3 and 8, whereas gene amplifications on chromosome 19 are mutually exclusive to BRCA mutations. Chromosome 19 amplifications include CCNE1 (19q12), BRD4 (19p13.1) and PAK4 (19q13.2) and are mainly identified in BRCA wild-type phenotype. BRD4 was amplified in 18% of BRCA wild-type tumors and in none of BRCA mutant-like tumors [19]. It is well-known that BRCA-mutated tumors are associated with a better prognosis. Consequently, the absence of BRCA1/2 mutations in BRD4 amplified tumors may affect prognosis. The presence of other mutations including BRCA1/2 that interfere with survival should be considered when exploring the association of BRD4 expression with survival. In addition, the prognostic role of BRD4 expression may require the simultaneous presence of other gene amplifications. One study of 579 patients with HGSC failed to prove the prognostic significance of BRD4 gene amplification and BRD4 protein overexpression alone [17]. However, the co-amplification of BRD4 and CCNE1 was associated with poor survival in HGSC patients. The amplification of either gene alone was not associated with disease outcome [17]. Indeed, CCNE1 amplification was also present in studies that showed that BRD4 amplification is a poor prognostic factor [18,22]. These data indicate that the overexpression of other genes other than BRD4 may interfere with study results and thus should be taken into account. 

Another cause of this contradiction over the prognostic role of BRD4 gene expression is the fact that it encodes two different isoforms: the long BRD4 isoform (BRD4-L) and the short BRD4 isoform (BRD4-S) [8,23]. Both BRD4 isoforms share identical N-terminal protein domain architecture, but they differ in their C-terminal amino acid residues because of alternative mRNA splicing. There are data that indicate a balanced, constant ratio of both isoforms is needed to ensure the homeostatic work of BRD4 protein [24]. The two BRD4 isoforms have opposing functions and the disruption of the balance between them may lead to significant biological consequences [23,24]. For instance, BRD4-S lacks the C-terminal domain and has stronger binding affinity to acetylated lysine residues than BRD4-L [24]. Other studies support that BRD4-S possesses an oncogenic activity while BRD4-L serves as a tumor suppressor [23]. Moreover, BRD4-S is shown to promote breast cancer growth and metastasis, while BRD4-L inhibits tumor progression and limits metastatic potential due to its suppressive role in growth factor-induced cell migration and cancer stem cell proliferation [23,24]. Of note, the two BRD4 isoforms are not inhibited equally by each inhibitor so if the inhibitor is more active against one isoform this may result in up-regulation of the other [25]. BRD4-S has been shown to serve as an endogenous inhibitor of DNA damage repair (DDR) as it binds stably to DNA and blocks its access by DDR complex. On the other hand, BRD4-L serves as a scaffold for DDR proteins in sites of DNA damage promoting non-homologous end joining (NHEJ) pathways [24]. It is now clear that each BRD4 isoform possesses a distinct role in carcinogenesis and disease prognosis. In our study there was no distinction between the two BRD4 isoforms and both were measured as a whole. It is important that each isoform is evaluated individually and that the prognostic role of each one is addressed separately. Indeed, we noticed an inverse association between BRD4 gene expression and protein level. BRD4-L transcript levels were inversely associated with protein level in a recent study. On the contrary, BRD4-S mRNA expression associates conversely with BRD4-S protein level [26]. Consequently, there is no clear association between BRD4 mRNA expression and protein level and this explains in part the nonlinear association presented in our study.

Posttranslational modifications of BRD4 protein are another issue that may affect our results. Firstly, BRD4 protein undergoes ubiquitination by E3 Ubiquitin ligase adaptor protein SPOP (Speckle-type POZ protein) that transfers the activated ubiquitin from E2 ubiquitin-conjugating enzyme and leads to proteasome degradation. SPOP gene mutations may increase or reduce sensitivity to BET inhibitors. Ubiquitination determines the stability of BRD4 protein and accumulation of BRD4 protein confers resistance to BET inhibitors [27]. In contrast, BRD4 polyubiquitination can be reversed through the activity of ubiquitin-specific processing protease 17 (USP17) also known as deubiquitinase 3 (DUB3). DUB3 binds to BRD4 protein and promotes its deubiquitination and stabilization. DUB3 stabilizes BRD4 through deubiquitination and promotes cancer progression. Another posttranslational modification of BRD4 is phosphorylation that is regulated by casein kinase II (CK2) and protein phosphatase 2A (PP2A). BRD4 protein contains two phosphorylation sites: one in the N-terminal phosphorylation site (NPS) of BD2 and another in the C-terminal phosphorylation site (CPS) [27]. The N-terminal domain is of high importance for BRD4 binding to acetylated histones and the recruitment of transcriptional factors to target genes. BRD4 phosphorylation by CK2 increases the stability and nuclear localization of BRD4 and thus its oncogenic activity. In addition, cyclin-dependent kinase 9 (CDK9) binds BRD4 through its C-terminal domain. BRD4 phosphorylation by CDK9 increases with the recruitment of P-TEFb by BRD4 as P-TEFb is the heterodimer of CDK9. Conversely, BRD4 dephosphorylation by protein phosphatase 2A (PP2A) impairs the association of BRD4 with acetylated nucleosomes and transcription factors. Downregulation of PP2A leads to BRD4 hyperphosphorylation and thus resistance to BET inhibitors. Overall, the inverse association between BRD4 gene and protein expression may be the result of these posttranslational modifications. Posttranslational modifications may contribute to the difference between the prognostic role of BRD4 gene and protein levels. For instance, BRD4 hyperphosphorylation and not overexpression has been associated with adverse prognosis in triple negative breast cancer [28].

Our study has certain strengths and limitations. A major strength is the analysis of BRD4 gene expression through RNA extraction from peripheral blood mononuclear cells (*PBMCs*) that were isolated from ascitic fluid. There have been many studies exploring the clinical role of the BRD4 gene and the effect of BET inhibitors in ovarian cancer cell lines but not in cells directly extracted from biological specimens. This provides a better insight on the in vivo function of the BRD4 gene in humans although the collection and processing of ascitic fluid can be rather challenging. One of the main limitations of the study is the small sample size. However, RNA and protein extraction from cells isolated from ascitic fluid is a demanding process. In addition, our study did not take into consideration the presence of other mutations, e.g., BRCA mutations, HRR gene alterations, that could affect prognosis. Finally, optimal debulking was not feasible in all the patients included even though all patients suffered from stage FIGO IIIC disease.

## 7. Conclusions

We conclude that intermediate/high BRD4 gene expression was associated with better prognosis in HGSC, while a BRD4 high protein level was inversely associated with prognosis although not significantly. Large studies are required to further assess the prognostic role of BRD4 in ovarian cancer.

## Figures and Tables

**Figure 1 cancers-16-01962-f001:**
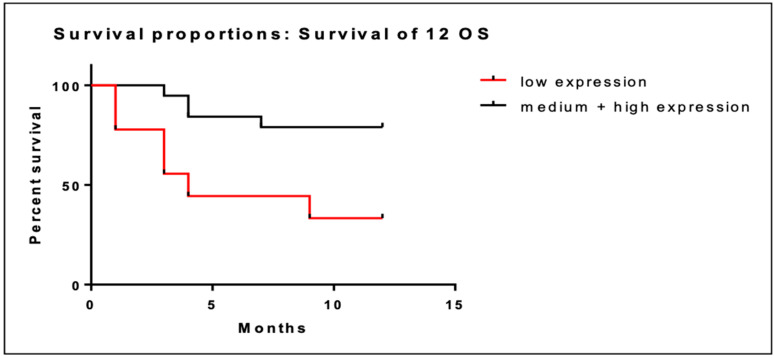
Kaplan–Meier curves of OS at 12 months of patients with low BRD4 expression (red line) versus intermediate/high gene expression (black line).

**Figure 2 cancers-16-01962-f002:**
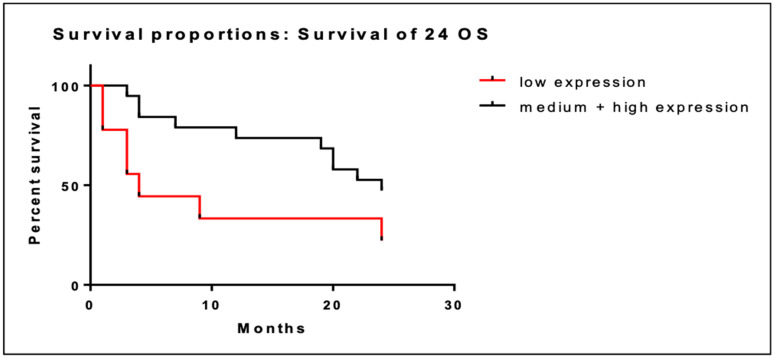
Kaplan–Meier curves of OS at 24 months of patients with low BRD4 expression (red line) versus intermediate/high gene expression (black line).

**Figure 3 cancers-16-01962-f003:**
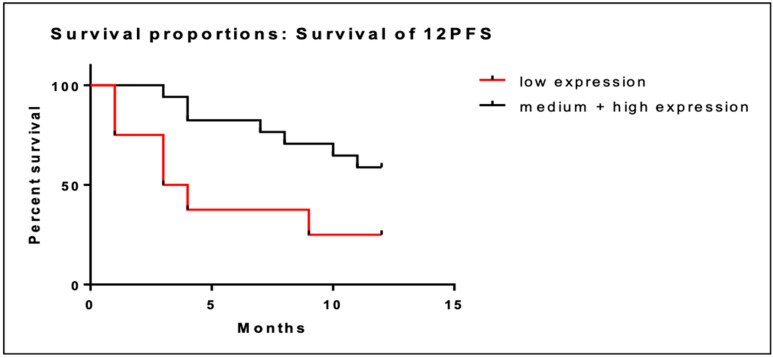
Kaplan–Meier curves of PFS at 12 months of patients with low BRD4 expression (red line) versus intermediate/high gene expression (black line).

**Figure 4 cancers-16-01962-f004:**
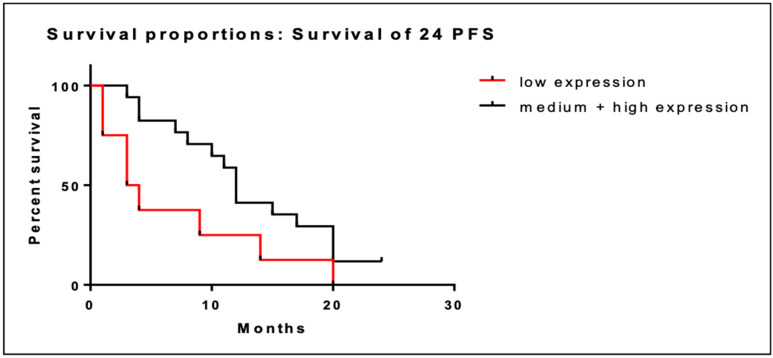
Kaplan–Meier curves of PFS at 24 months of patients with low BRD4 expression (red line) versus intermediate/high gene expression (black line).

**Figure 5 cancers-16-01962-f005:**
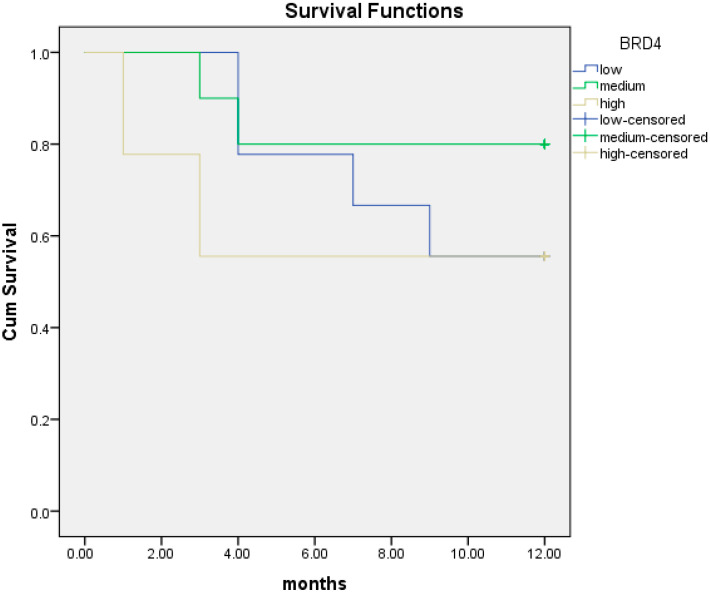
Kaplan–Meier curves of overall survival (OS) at 12 months in patients with low (blue line), intermediate (green line) and high (yellow line) BRD4 protein expression.

**Figure 6 cancers-16-01962-f006:**
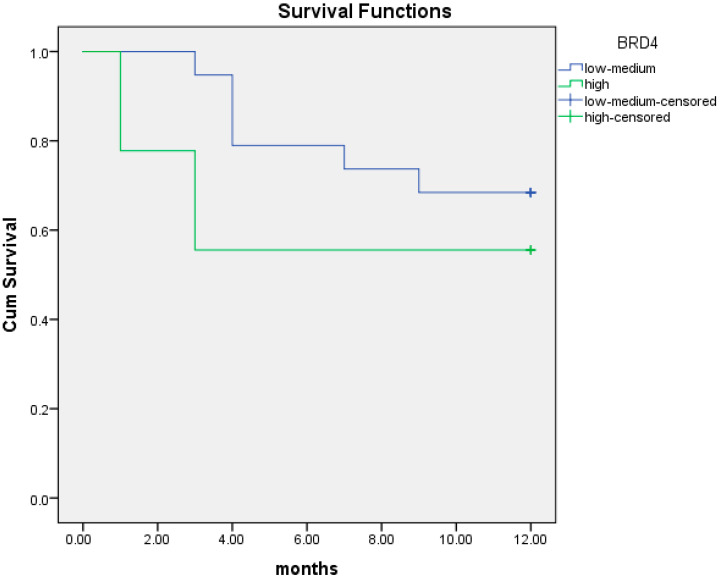
Kaplan–Meier curves of overall survival (OS) at 12 months in patients with intermediate/low (blue line) and high (green line) BRD4 protein expression.

**Figure 7 cancers-16-01962-f007:**
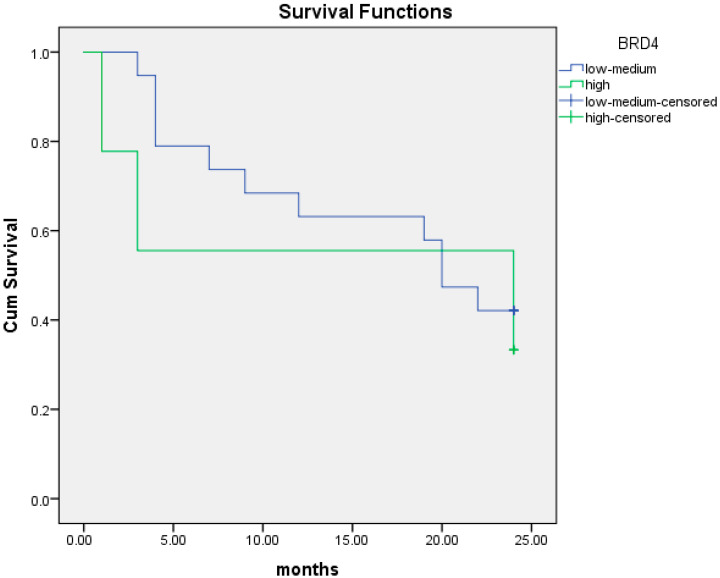
Kaplan–Meier curves of overall survival (OS) at 24 months in patients with intermediate/low (blue line) and high (green line) BRD4 protein expression.

**Figure 8 cancers-16-01962-f008:**
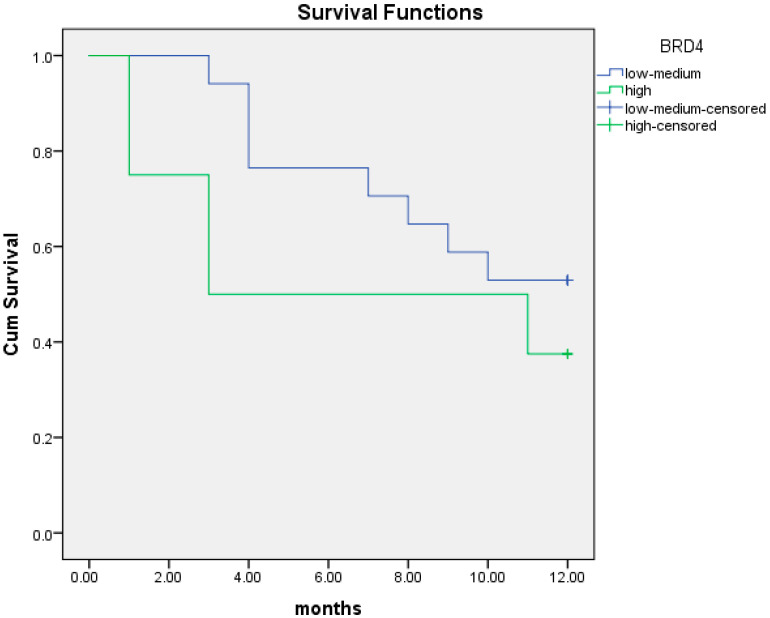
Kaplan–Meier curves of PFS at 12 months in patients with intermediate/low (blue line) and high (green line) BRD4 protein expression.

**Figure 9 cancers-16-01962-f009:**
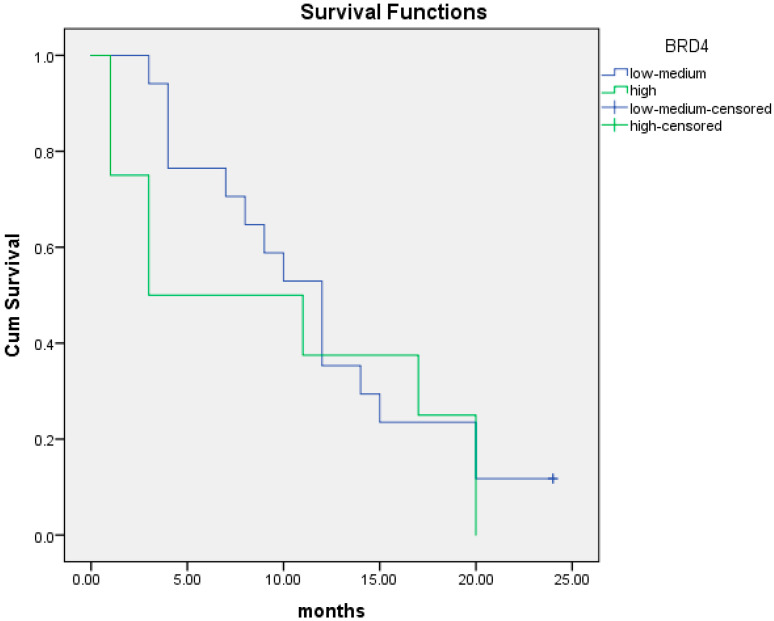
Kaplan–Meier curves of PFS at 24 months in patients with intermediate/low (blue line) and high (green line) BRD4 protein expression.

**Figure 10 cancers-16-01962-f010:**
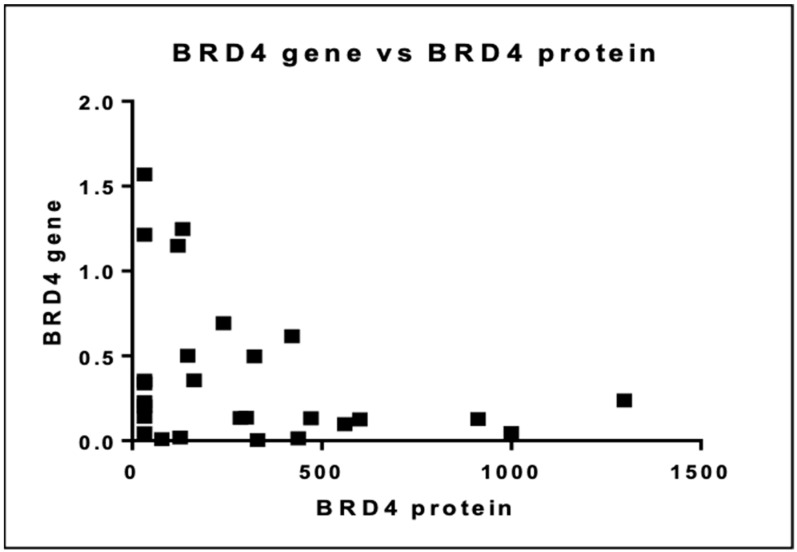
Association between BRD4 gene and protein expression.

**Table 1 cancers-16-01962-t001:** Clinicopathological characteristics of the patients.

Characteristic	Total N = 28, *n* (%)
**Age at diagnosis, median (range), years**	66 (42–86)
**Histology**	
**High-Grade Serous carcinoma**	28 (100%)
**Other**	0 (0)
**Initial Stage (FIGO)**	
**IIIC**	28 (100%)
**IV**	0 (0)
**Family History**	
**Yes**	7 (25%)
**No**	21 (75%)
**Debulking Surgery**	
**Primary Debulking Surgery**	10 (36%)
**Interval Debulking Surgery**	12 (43%)
**No surgery**	6 (21%)
**ECOG Performance status**	
**0/1**	25 (89%)
**2/3**	3 (11%)
**Progression-free survival on first-line chemotherapy, median (range), months**	11 (1–33)
**Sites at relapse**	
**Peritoneum**	20 (71%)
**Lymph node**	11 (39%)
**Mediastinum**	0 (0)
**Other**	7 (25)
**Prior bevacizumab treatment**	12 (43%)
**Prior PARP inhibitor treatment**	18 (64%)
**BRCA mutations**	
**BRCA1 somatic**	5 (18%)
**BRCA2 somatic**	1 (4%)
**Wild-type**	22 (78%)

**Table 2 cancers-16-01962-t002:** Overall survival at 12 months according to BRD4 gene expression.

Means and Medians for Survival Time								
BRD4	Mean				Median			
	Estimate	Std. Error	95% Confidence Interval		Estimate	Std. Error	95% Confidence Interval	
			Lower Bound	Upper Bound			Lower Bound	Upper Bound
Low	6.333	1.523	3.347	9.319	4.000	1.491	1.078	6.922
Medium High	10.421	0.719	9.012	11.830				
Overall	9.107	0.780	7.579	10.635				

## Data Availability

Data presented in our study can be found in the patients’ archive that is safely stored in our institution. The patients’ archive is available from the corresponding author upon request.

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
