# Peer review of "The Prognostic Role of BRD4 Expression in High-Grade Serous Ovarian Cancer"

_cancers, 2024, doi:10.3390/cancers16111962_

Round 1

Reviewer 1 Report

Comments and Suggestions for Authors

The manuscript titled The Prognostic Role Of BRD4 Expression in High-Grade 2 Serous Ovarian Cancer by Angeliki Andrikopoulou et al. I have the following suggestions.

- Since the BRD4 gene expression and protein levels are not correlated to each other, I wonder what’s the underlying mechanism.

- The ethnicity of the patients enrolled in this study should be mentioned.

- Some of the figures should be improved and presented in higher resolution.

- Numbers and units should have a space between them, e.g.  “10pmol, 10mM, 200nM, 75ng”, should be “10 pmol, 10 mM, 200 nM, 75 ng”

- Tables should be shown as one whole table where possible, e.g. Table 2, for the ease of reading.

- I am a bit confused in the abstract and the conclusion, as there is contradictory wordings in these two sections on BRD4 expression.

- Typos and unfriendly mode of English usage can be found.

Comments on the Quality of English Language

none

Author Response

Athens, May 2024            

Re: Rebuttal letter to Reviewers’ comments on Manuscript: cancers-2998584

(Submission title: “The Prognostic Role of BRD4 Expression in High-Grade Serous Ovarian Cancer”)

Dear Editor,

Following your email from May 7, 2024, we are submitting the revised version of our manuscript for consideration. We would like to thank the Editor and the Reviewers for carefully reviewing our work and for the very thoughtful suggestions. Please find below our point-by-point answers to reviewer’s comments. We have numbered the lines and highlighted the changes in background yellow color for your further convenience.

REVIEWER 1

The manuscript titled The Prognostic Role Of BRD4 Expression in High-Grade 2 Serous Ovarian Cancerby Angeliki Andrikopoulou et al. I have the following suggestions.

- Since the BRD4 gene expression and protein levels are not correlated to each other, I wonder what’s the underlying mechanism.

We would like to thank the Reviewer for his/her thoughtful comment and for giving us the opportunity to explain this. Indeed, this observation is what troubled us the most. This is why we searched the literature and we have provided many reasons why this could be observed – nearly half of the Discussion section is devoted to this observation (Lines 341-395). Some of the possible explanations include:
  1. BRD4 gene encodes two different isoforms of BRD4 protein with opposing functions: the long BRD4 isoform (BRD4-L) and the short BRD4 isoform (BRD4-S). We do not know which one was measured in our study as there was no distinction between them. For instance, if we measured BRD4-S that possesses oncogenic activity our findings come in agreement with the expected ones.
  2. ubiquitination of BRD4 protein by E3 Ubiquitin ligase adaptor protein SPOP (Speckle-Typepoz protein) that leads to proteasome degradation irrespective of the initial BRD4 gene expression.
  3. stabilization of BRD4 protein by deubiquitination by deubiquitinase 3 (DUB3)
  4. phosphorylation and dephosphorylation of BRD4 protein by casein kinase II (CK2) and protein phosphatase 2A (PP2A) respectively that determines its function.
Finally, we here highlight that there is no linear association between BRD4 protein and gene expression so we found no correlation between them. 

- The ethnicity of the patients enrolled in this study should be mentioned.

We thank the Reviewer for pointing this out – we now provide the ethnicity in Line 107.

- Some of the figures should be improved and presented in higher resolution.

We agree with the Reviewer, and we now upload the Figures in higher resolution.

- Numbers and units should have a space between them, e.g.  “10pmol, 10mM, 200nM, 75ng”, should be “10 pmol, 10 mM, 200 nM, 75 ng”

We have now modified the manuscript accordingly as seen in Lines 118, 145-146, 151, 154.

- Tables should be shown as one whole table where possible, e.g. Table 2, for the ease of reading.

We thank the Reviewer for his/her suggestion – all Tables are separated as they present different results. However, the Editor can edit Tables according to the Journal’s specific requirements.

- I am a bit confused in the abstract and the conclusion, as there is contradictory wordings in these two sections on BRD4 expression.

We totally agree with the reviewer – we have modified the conclusion section accordingly (Line 409) so that there is no confusion.

- Typos and unfriendly mode of English usage can be found.

We would like to thank the Reviewer for pointing this out – our manuscript was reviewed by a native English speaker and many parts have been corrected (Lines 57, 59, 72-74, 81-85, 100, 192, 299, 344, 372).

We would like to thank once more the editor and the reviewers for their careful and thorough reading of the manuscript.

Angeliki Andrikopoulou

MD, PhD

National and Kapodistrian University of Athens

Reviewer 2 Report

Comments and Suggestions for Authors

This study investigated the relationship between BRD4 gene expression levels and both overall survival (OS) and progression-free survival (PFS), along with BRD4 protein expression and its impact on OS and PFS at 12 and 24 months. The findings suggested that lower BRD4 gene expression correlated with a more favorable prognosis, while higher levels of BRD4 protein were tentatively associated with worse outcomes, though not statistically significant. These results hold significant implications for clinical practice. However, I still have one question. What are the method and criteria for dividing the patients to 3 group by BRND4 gene and protein levels. Please provide more details.

Author Response

Athens, May 2024            

Re: Rebuttal letter to Reviewers’ comments on Manuscript: cancers-2998584

(Submission title: “The Prognostic Role of BRD4 Expression in High-Grade Serous Ovarian Cancer”)

Dear Editor,

Following your email from May 7, 2024, we are submitting the revised version of our manuscript for consideration. We would like to thank the Editor and the Reviewers for carefully reviewing our work and for the very thoughtful suggestions. Please find below our point-by-point answers to reviewer’s comments. We have numbered the lines and highlighted the changes in background yellow color for your further convenience.

REVIEWER 2

This study investigated the relationship between BRD4 gene expression levels and both overall survival (OS) and progression-free survival (PFS), along with BRD4 protein expression and its impact on OS and PFS at 12 and 24 months. The findings suggested that lower BRD4 gene expression correlated with a more favorable prognosis, while higher levels of BRD4 protein were tentatively associated with worse outcomes, though not statistically significant. These results hold significant implications for clinical practice. However, I still have one question. What are the method and criteria for dividing the patients to 3 group by BRND4 gene and protein levels. Please provide more details.

We would like to thank the Reviewer for his/ her valuable guidance on improving our manuscript. We performed an initial quantification of the BRD4 gene expression as it was measured through RT-qPCR. Median expression of BRD4 gene was 0.170 – however we wanted to separate patients in three groups each one consisting of equal number of patients. The upper limit of those with LOW BRD4 gene expression was 0.130 (9 patients), MEDIUM expression was 0,130 – 0,354 (10 patients) and HIGH expression was 0,354 - 1,56 (9 patients). This allowed us to divide the whole population in three subgroups consisting of EQUAL number of patients.

Respectively, median BRD4 protein level as measured via ELISA was 201.4 pg/ml. We separated patients on the same way for BRD4 protein as well: 9 patients with low BRD4 protein level < 102 pg/ml, 10 patients with medium BRD4 protein level (102 - 323 pg/ml) and 9 patients with high BRD4 protein level (> 323 – 1298). We performed our analysis on these subgroups to distribute all patients equally. We would like to thank the Reviewer for giving us the opportunity to explain this.       

We would like to thank once more the editor and the reviewers for their careful and thorough reading of the manuscript.

Angeliki Andrikopoulou

MD, PhD

National and Kapodistrian University of Athens

Round 2

Reviewer 1 Report

Comments and Suggestions for Authors

The authors have addressed most of the concerns.